# Antifungal and Herbicidal Potential of *Piper* Essential Oils from the Peruvian Amazonia

**DOI:** 10.3390/plants11141793

**Published:** 2022-07-07

**Authors:** Liliana Ruiz-Vásquez, Lastenia Ruiz Mesia, Henrry Denny Caballero Ceferino, Wilfredo Ruiz Mesia, Maria Fe Andrés, Carmen Elisa Díaz, Azucena Gonzalez-Coloma

**Affiliations:** 1Laboratorio de Productos Naturales Antiparasitarios de la Amazonia, Centro de Investigación de Recursos Naturales, Universidad Nacional de la Amazonia Peruana (UNAP), Iquitos 16002, Peru; lastenia.ruiz@unapiquitos.edu.pe (L.R.M.); henrrycaballero.1991@gmail.com (H.D.C.C.); wilfredo.ruiz@unapiquitos.edu.pe (W.R.M.); 2Facultad de Farmacia y Bioquímica, Universidad Nacional de la Amazonia Peruana (UNAP), Iquitos 16000, Peru; 3Instituto de Ciencias Agrarias, CSIC, 28006 Madrid, Spain; mafay@ica.csic.es; 4Instituto de Productos Naturales y Agrobiología, CSIC, 38206 La Laguna, Spain; celisa@ipna.csic.es

**Keywords:** *Piper*, essential oil, chemical composition, antifungal and phytotoxic

## Abstract

The chemical composition of essential oils (EOs) from ten Peruvian *Piper* species (*Piper coruscans*, Pc; *P. tuberculatum*, Pt; *P. casapiense*, Pcs; *P. obliquum*, Po; *P. dumosum*, Pd; *P. anonifolium*, Pa; *P. reticulatum*, Pr; *P. soledadense*, Ps; *P. sancti-felicis*, Psf and *P. mituense*, Pm) has been studied, along with their antifungal and phytotoxic activities. These EOs contained β-bisabolene/nerolidol (Pc), β-bisabolene/δ-cadinene/caryophyllene (Pt), caryophyllene oxide (Pcs), bicyclogermacrene/10-epi-Elemol (Po), bicyclogermacrene/germacrene-D/apiol (Pd), caryophyllene/germacrene-D (Pa), germacrene-D (Pr), limonene/apiol (Ps), apiol (Psf), and apiol/bicyclogermacrene (Pm) as major components, and some are described here for the first time (Ps, Pcs, Pm). A composition-based dendrogram of these *Piper* species showed four major groups (G1: Pc and Pt, G2: Pcs, Po, Pd, Pa, and Pr, G3: Ps, and G4: Psf and Pm). The spore germination effects (*Aspergillus niger*, *Botrytis cinerea*, and *Alternaria alternate*) and phytotoxicity (*Lolium perenne* and *Lactuca sativa*) of these EOs were studied. Most of these *Piper* essential oils showed important activity against phytopathogenic fungi (except G1), especially against *B. cinerea*. Similarly, most of the essential oils were phytotoxic against *L. perenne* (except G1), with *P. sancti-felicis* (G4), *P. casapiense* (G2), and *P. reticulatum* (G2) being the most effective. Caryophyllene oxide, β-caryophyllene, β-pinene, limonene, α-humulene, and apiol were evaluated against *B. cinerea*, with the most effective compounds being β-pinene, apiol, and limonene. This work demonstrates the species-dependent potential of essential oils from Peruvian *Piper* species as fungicidal and herbicidal agents.

## 1. Introduction

Diseases caused by plant pathogens significantly contribute to annual loss in crop yield worldwide [1]. The application of chemical pesticides may be an important component and effective control method of these plant diseases, both representing a serious threat to public health and the environment while also causing resistance in pathogens [2,3,4,5,6].

In plants, essential oils (EOs) play a protective role against herbivores, phytopathogenic fungi, and weeds. Essential oils also represent a new class of crop protectants due to their volatility and low toxicity to the environment and have been proposed as an alternative to synthetic pesticides [3,7,8,9]. Additionally, the probabilities of creating new resistant pathogens by using essential oils as biopesticide agents are low, since their constituents can act as synergists [10].

The Piperaceae family has approximately eight genera and 3000 species [11]. The genus *Piper* is found in tropical and subtropical areas, and in America, there are approximately 700 species [12,13], with 324 species located in Peru [14,15]. This genus is an important source of essential oils and secondary metabolites, which have significant plant protection effects [16], including allelopathic/phytotoxic [14,17,18], antifungal [19,20], insecticidal, nematicidal, and antifeedant [10,21].

*Piper* EOs are characterized by the presence of monoterpene hydrocarbons (e.g., α-pinene, myrcene, limonene, α-terpinene), oxygenated monoterpenoids (e.g., linalool, 1,8-cineole, terpinen-4-ol, borneol), sesquiterpene hydrocarbons (e.g., β-caryophyllene, α-humulene, germacrene D, bicyclogermacrene, α-cubebene), oxygenated sesquiterpenoids (e.g., spathulenol, (*E*)-nerolidol, caryophyllene oxide, α-cadinol, epi-α-bisabolol), and phenylpropanoids (safrole, dillapiol, myristicin, elemicin, apiol, eugenol), among others [14,22,23,24]. Furthermore, *Piper* essential oils have been described as being insect antifeedant, acaricidal, nematicidal, and herbicidal agents [10,21]. Therefore, *Piper* essential oils are a promising source of new potential biopesticide ingredients.

As part of an ongoing project on the bioprospection of Peruvian *Piper* species for their biopesticidal potential, ten species native to the Peruvian Amazonian region (*P. coruscans, P. tuberculatum, P. casapiense, P. obliquum, P. dumosum, P. anonifolium, P. reticulatum, P. soledadense, P. sancti-felicis*, and *P. mituense*) have been extracted by hydrodistillation to study the chemical composition of their EOs by GC-MS, along with their fungicidal activity against phytopathogens (*Aspergillus niger, Botrytis cinerea*, and *Alternaria alternate*) and their phytotoxic effects (against *Lolium perenne* and *Lactuca sativa*) to assess their potential applications in phytopathogen and/or weed control.

## 2. Results

### 2.1. Essential Oil Composition

The chemical compositions of the essential oils are shown in Table 1 and the structures of their main components in Figure 1. The major identified compounds of the EOs were β-bisabolene (33.4%), nerolidol (10.2%), caryophyllene (8.0%), (+)-β-selinene (4.9%), and α-bisabolol (4.8%) for *Piper coruscans;* β-bisabolene (40.2%), δ-cadinene (9.8%), caryophyllene (9.7%), germacrene-D (5.0%), and nerolidol (4.5%) for *P. tuberculatum;* caryophyllene oxide (10.2%), and caryophyllene (4.7%) for *P. casapiense*; bicyclogermacrene (7.9%), 10-epi-Elemol (7.3%), caryophyllene (6.3%), α-pinene (6.0%), β-pinene (5.1%), β-selinenol (4.9%), α-eudesmol (4.5%), and camphene (4.4%) for *P. oblicuum*; bicyclogermacrene (16.5%), germacrene-D (10.4%), apiol (8.9%), caryophyllene (6.8%), β-pinene (6.3%), α-cubebene (5.9%), and β-elemene (4.5%) for *P. dumosum*; caryophyllene (11.3%), germacrene-D (9.6%), α-humulene (6.6%), δ-cadinene (6.6%), and (-)-β-copaene (5.8%) for *P. anonifolium*; germacrene-D (12.6%), bicyclogermacrene (8.1%), δ-cadinene (6.0%), copaene (4.6%), and caryophyllene (4.5%) for *P. reticulatum*; limonene (38.5%), apiol (15.0%), caryophyllene oxide (8.4%), eudesma-3,7-(11)-diene and copaene (5.8%) for *P. soledadense*; apiol (76.1%), and caryophyllene (4.1%) for *P. sancti-felicis* and apiol (51.6%), bicyclogermacrene (9.0%), germacrene-D (6.7%), and myristicin (4.6%) for *P. mituense*.

The overall composition of these oils is shown in Table 2. Sesquiterpene hydrocarbons were dominant in all EOs except for *P. sancti-felicis* and *P. mituense*, which were characterized by phenylpropanoids, and *P. soledadense*, which was characterized by monoterpene hydrocarbons.

A dendrogram based on the composition of the *Piper* species (Figure 2) showed four groups: (G1) *P. coruscans* (Pc) and *P. tuberculatum* (Pt), characterized by the presence of sesquiterpene hydrocarbons; (G2) *P. casapiense* (Pcs), *P. obliquum* (Po), *P. dumosum* (Pd), *P. anonifolium* (Pa), and *P. reticulatum* (Pr), characterized by sesquiterpenes; (G3) *P. soledadense* (Ps), with monoterpenes and sesquiterpenes; and (G4) *P. sancti-felicis* (Psf) and *P. mituense* (Pm), characterized by phenylpropanoids.

### 2.2. Fungicidal Activity

The antifungal activity (spore germination inhibition) of the *Piper* essential oils against *Aspergillus niger, Botrytis cinerea*, and *Alternaria alternate* is shown in Table 3. *B. cinerea* was the fungal species most susceptible to the action of *Piper* essential oils. The antifungal activity showed a pattern in accordance with the grouped EOs. *P. coruscans* and *P. tuberculatum* (G1) were not active. P. *casapiense* (G2), *P. obliquum* (G2), *P. dumosum* (G2), *P. anonifolium* (G2), and *P. reticulatum* (G2) were only active against *B. cinerea* with varying potencies, with *P. obliquum* and *P. anonifolium* being the most effective. *P. soledadense* (G3) inhibited the spore germination of *A. niger* and *B. cinerea*. *P. sancti-felicis* (G4) oil was active against all three fungal species with moderate effects, and *P. mituense* (G4) only acted on *B. cinerea*, probably due to the lower concentration in apiol of this oil.

Among the *Piper* oils components tested against *B. cinerea* (β-pinene, limonene, α-humulene, β-caryophyllene, caryophyllene oxide and apiol), β-pinene showed strong antifungal activity (CI_50_ = 3.48 µg/mL, 2.69–4.50 95% CL), followed by apiol (CI_50_ = 16.17 µg/mL, 10.74–24.34 95% CL) and limonene (CI_50_ = 34.70 µg/mL, 24.38–49.4 95% CL), with effective doses similar to the positive control (thymol, CI_50_ = 19.54 µg/mL, 22.94–15.74 95% CL) for apiol.

### 2.3. Phytotoxic Activity

The essential oils were tested for phytotoxic effects on *Lactuca sativa* (dicotyledonous) and *Lolium perenne* (monocotyledonous) plants. The phytotoxic activity did not follow the grouping pattern observed for the antifungal effects. *P. sancti-felicis* (G4), *P. casapiense* (G2), *P. mituense* (G4), and *P. reticulatum* (G2), effectively inhibited germination, leaf and root growth of *L. perenne* (>50%, Figure 3). *P. anonifolium* (G2), *P. obliquum* (G2), *P. dumosum* (G2), and *P. soledadense* (G3) inhibited leaf growth (>50%), followed by *P. tuberculatum* (G1) with a 50% inhibition.

*P. sancti-felicis, P. casapiense, P. reticulatum*, and *P. mituense* effectively inhibited the root growth of *L. sativa* (data not shown). *P. soledadense* reduced the root growth of *L. sativa* (data not shown). These results show strong selective herbicidal potential of the EOs tested against monocotyledonous plants.

## 3. Discussion

*Piper* essential oils present a wide variety of chemical compounds with important biological activities that may be of interest in agriculture, medicine, and food industries, among others. These oils play an important role in the defense of the plant against pests, and many studies have reported activities as insecticidal, antifeedants, phytotoxic and antifungal [10,19,20,21,25,26,27,28].

The essential oils from some of the *Piper* species described here have been previously reported to show quantitative and qualitative chemical variations that can be attributed to environmental factors (such as weather, soil, sunlight, temperature, and humidity) [13,29,30]. Among the essential oils studied here, four groups have been identified according to their compositions (G1–G4):

(G1) *P. coruscans* (Pc) and *P. tuberculatum* (Pt). *P. coruscans* EO had β-bisabolene (33.4%) and nerolidol (10.2%) as the main components, while this species collected in Ecuador contained β-caryophyllene (24.1–25.0%), α-humulene (11.6–12.0%), and caryophyllene oxide (9.3–10.9%) [31]. The EO from *P. tuberculatum* studied here showed β-bisabolene (40.2%) as the main component followed by δ-cadinene (9.8%), caryophyllene (9.7%), germacrene-D (5.0%), nerolidol (4.5%), copaene (4.2%), and β-elemene (3.3%). This *Piper* species (Pt) EO has been previously reported for plants collected from different locations. Pt collected in Venezuela gave an EO with α-farnesene (6.2%), humulene epoxide II (6.0%), 2-pentadecanone (4.1%), β-eudesmol (4.4%), 2-tridecanone (4.3%), ledane (3.6%), (*E,E*)-farnesylacetone (3.6%), and α-cadinol (2.9%) [32], while EOs from the Brazilian Amazonia or Mato Grosso regions had either (*E*)-caryophyllene (30.1%) [19] or myristicin (15.5%), dillapiole (13.8%), α-guaiene (13.0%), 9-epi-*E*-cariofilene (7.1%), and *trans*-4-muurola(14)-5-diene (9.9%) [33].

(G2) *P. casapiense* (Pcs), *P. obliquum* (Po), *P. dumosum* (Pd), *P. anonifolium* (Pa), and *P. reticulatum* (Pr): *P. casapiense* (Pcs), is reported here for the first time. The EO from *P. obliquum* had bicyclogermacrene (7.9%), 10-epi-Elemol (7.3%), caryophyllene (6.3%), and α-pinene (6.0%). However, the Po essential oil of Ecuadorian origin contained safrole (45.9%), γ-terpinene (17.1%), and terpinolene (11.5%) [34], while an EO from Panama had β-caryophyllene (27.6%), spathulenol (10.6%), and caryophyllene oxide as main components (8.3%) [35]. *P. dumosum* contained bicyclogermacrene (16.5%) and germacrene-D (10.4%). Similarly, the EO from Pd plants collected in the Brazilian Amazonia had bicyclogermacrene (16.2%), β-caryophyllene (15.9%), β-pinene (16.0%), and α-pinene (12.1%) [36] as the main components. The EO from *P. anonifolium* studied here contained caryophyllene (11.3%), germacrene-D (9.6%), δ-cadinene (6.6%), α-humulene (6.6%), and neoalloocimene (5.5%), while a previously reported EO from the Brazilian Pará was composed of selin-11-en-4-β-α-ol (20.0%), β-selinene (12.7%), α-selinene (11.9%), and α-pinene (8.8%) [37]. Furthermore, the EOs from seven Brazilian Amazonian populations of Pa showed two different chemotypes, one rich in α-pinene (40.9–53.1%)/β-pinene (17.2–22.9%), and one rich in α-eudesmol (33.5%)/ishwarane (19.1%) [38]. The EO from *P. reticulatum* studied here contained the phenylpropanoid apiol as the main component (15.0%) while β-elemene (24.6%) and β-caryophyllene (16.7%) were abundant in a Pr oil from the northern region of Brazil [39].

(G3) P. *soledadense* (Ps) is reported here for the first time.

(G4) *P. sancti-felicis* (Psf) and *P. mituense* (Pm). The EO from *P. sancti-felicis* and *P. mituense* (Pm reported here for the first time) contained the phenylpropanoid apiol as the main component (76.1% and 51.6%, respectively). *δ-3*-carene (35.3%) and limonene (27.1%) were the main components of the EO from *P. sancti-felicis* collected in Choco, Colombia [21,25].

The essential oils studied here have shown important activities against phytopathogenic fungi (G2-4). Essential oils from the genus *Piper* have been reported to have a wide range of biological properties [10,11,14,21,40], including fungicidal effects [35,38,39]. Specifically, among the species studied here, the essential oils of *P. tuberculatum* (rich in α- and β-pinene 17–27%, (E)-β-ocimene 14% and β-caryophyllene 32.1%) [41], and *P. anonifolium* (with selin-11-en-4α-ol 20.0%, β-selinene 12.7%, and α-selinene 11.9%) showed strong antifungal activity against *Cladosporium cladosporioides* and *C. sphaerospermum* [37].

The composition-based grouping of the EOs overlapped with the antifungal activity, suggesting that the presence of bicyclogermacrene, 10-epi-Elemol, germacrene-D, caryophyllene, limonene, β-pinene, and/or apiol could be responsible for significant antifungal effects. Among the oil components tested against *B. cinerea* (β-pinene, limonene, α-humulene, β-caryophyllene, caryophyllene oxide, and apiol), β-pinene, limonene, and apiol showed strong antifungal activity. Therefore, β-pinene could contribute to the activity of Po and Pd EOs, while apiol and limonene could explain the effect of Ps, Psf, and Pm oils. β-caryophyllene (inactive) could be a synergist.

(S)-Limonene has reported antifungal activity against *Rhizoctonia solani*, *Fusarium oxysporum*, *Penicillium digitatum* and *Asperigallus niger* [42]. Apiol showed strong activity against *Botrytis cinerea* [43], *Aspergillus flavus*, *A. niger*, *A. fumigatus*, and *A. parasiticus* [44], *Botryodiplodia theobromae* and *Colletotrichum acutatum* [45,46]. The antifungal activity of apiol has been attributed to the presence of two electron-donating methoxy groups [47]. Caryophyllene oxide was active against *Fusarium solani* [48] and caryophyllene was active against *Rhizoctonia solani* and *Helminthosporium oryzae* [49], but in this work did not inhibit *B. cinerea* spore germination.

Most of the essential oils tested here (*P. sancti-felicis*, *P. mituense*, *P. casapiense*, *P. reticulatum*, *P. anonifolium, P. obliquum, P. dumosum*, and *P. soledadense*) were phytotoxic to the monocotyledonous *Lolium perenne*, and this activity did not overlap with the composition-based groups, suggesting a multi-component phytotoxic action. This is the first report on the phytotoxic effects of these species, except for an EO from *P. sancti-felicis* that was not active and did not contain apiol [21]. EOs from other *Piper* species, including *P. hispidinervum* rich in safrole [10], *P. dilatatum* rich in apiol, and *P. divaricatum* rich in eugenol and methyleugenol [21], have been reported as being phytotoxic, suggesting that the presence of phenylpropanoids played an important role in their activity.

Among the main compounds present in the *Piper* oils studied here, α-pinene, β-pinene, and limonene have been reported to have phytotoxic effects against the germination and seedling growth of several plant species [50], apiol inhibited the growth of *Lemna paucicostata* and was toxic against *Festuca rubra* and *Agrostis stolonifera* (monocot plants) [43], and caryophyllene inhibited the germination and seedling growth of *Brassica campestris* and *Raphanus sativus* [51] and the seed germination and root growth of *Echinochloa crusgalli, Lolium perenne, Amaranthus retroflexus*, and *Digitaria sanguinalis* [52].

## 4. Materials and Methods

### 4.1. Plant Material and Essential Oil Extraction

The aerial parts of the plants (leaves, stems, and flowers) of the selected Piperaceae species were collected in Iquitos, Loreto Department, Peru in different seasons. The taxonomic identification was carried out at the Herbarium Amazonense of the National University of the Peruvian Amazon, Iquitos, Peru. A voucher for each species has been deposited in the herbarium. All the plants were permitted for collection (Regional Management Resolution number 035-2021-GRL-GGR-GRDFFS). The EOs extraction was performed by hydrodistillation using the dried aerial parts of the plants. The EOs were separated by decantation and dried over anhydrous Na_2_SO_4_. All investigated *Piper* species contained essential oils that range from 0.078 to 1.26% based on dry weight (Table 4).

### 4.2. Gas Chromatography Analysis

Essential oils were analyzed by gas chromatography (GC) on a Shimadzu 2010 and gas chromatography-mass spectrometry (GC-MS) equipped with a mass spectrometer Shimadzu GCMS-QP2010-Ultra Mass Detector (electron ionization, 70 eV, Kyoto, Japan). The carrier gas was helium. The capillary column was a Teknokroma TRB (95%) dimethyl (5%) dimethylpolysiloxane (30 m × 0.25 mm ID and 0.25 µm phase thickness). Working conditions were as follows: injector temperature, 300 °C; column temperature 70–290 °C, for 6 min, staying at 290 °C for 15 min, temperature of the transfer line connected to the mass spectrometer, 250 °C, and ionization source temperature 250 °C. The identification of compounds was performed with standard terpenes analyzed under the same conditions and by comparison of the mass spectra with those available in the library Wiley Mass Spectral Database (Wiley 275 Mass Spectra Database, 2001), while relative area% has been used for quantification of all the peaks obtained in the chromatograms. The mass spectra and Kovats retention indexes obtained were compared with the literature reported [53,54].

### 4.3. Spore Germination Inhibition Assay

The fungal species *Aspergillus niger, Alternaria alternata*, and *Botrytis cinerea* came from the fungal collection of Instituto de Ciencias Agrarias-CSIC, Madrid, Spain where they are maintained. The antifungal activity of the essential oils was determined using a modified spore germination inhibition growth assay [55]. The essential oils (800, 400, 200, 100, and 50 µg/mL), and pure compounds (100, 500, 25, 12.5, 6.25, 3.125, and 1.56 µg/mL) were dissolved in dimethyl sulfoxide (DMSO) at 1% and evaluated at the final concentrations indicated. The spore suspensions were 7.5 × 10^5^ cells/mL in NaCl 0.9% for *A. niger* and 1 × 10^7^ cells/mL in distilled water for *B. cinerea* and *A. alternate*. Amphotericin B (5 µg/mL) was used as a positive control.

The samples and spore suspensions (4 replicates) were placed on 96-well plates and incubated for 24 h (28 °C for *A. niger* and 25 °C for *B. cinerea*). After the incubation process, 25 uL of an MTT (5 mg/mL) plus menadione (1 mM) solution in RMPI_MOPS_ were added, the plates were incubated again for 3 h, the medium was removed, 200 µL of acidic isopropanol (95% isopropanol and 5% 1 M HCl) was added, and the plates were incubated for another 30 min. The absorbance was read at 490 nm in an Elisa reader. The IC_50_ values (the effective dose to give 50% inhibition) were calculated by a regression curve of % spore germination inhibition on log dose.

The pure compounds β-pinene, limonene, α-humulene, β-caryophyllene, caryophyllene oxide, and apiol were from Sigma Aldrich (St. Louis, MO, USA).

### 4.4. Phytotoxic Activity

These experiments were conducted with *Lactuca sativa*, and *Lolium perenne* seeds (40 seeds/test) in 12-well microplates, as described previously [56]. The essential oils were tested at initial concentrations of 0.2 mg/mL (final concentration in the well), respectively. Juglone (Sigma) was included as positive control (0.1 mg/mL), resulting in 100% germination inhibition. Germination was monitored for six (*L. sativa*) or seven days (*L. perenne*), and the root length (25 plants randomly selected and digitalized) was measured (ImageJ, http//rsb.info. nih.gov./ij/; accessed on 20 February 2021) at the end of the experiment. A nonparametric analysis of variance (ANOVA) was performed on root/leaf length data [10,21].

### 4.5. Statistical Analysis

The data were analyzed using STATGRAPHICS Centurion XIX (https://www.statgraphics.com, accessed on 2 July 2022).

The variability of the chemical composition of each oil was assessed based on relative concentration data (% composition values for each species) subjected to cluster analysis (Farthest Neighbor Method, Squared Euclidean). The groups were chosen with a distance >2.

## 5. Conclusions

This work demonstrates the species-dependent potential of essential oils from Peruvian *Piper* species as fungicidal and herbicidal agents based on their composition. A dendrogram based on the composition of the *Piper* species showed four groups: (G1) *P. coruscans* (Pc) and *P. tuberculatum* (Pt), characterized by the presence of sesquiteterpene hydrocarbons; (G2) *P. casapiense* (Pcs), *P. obliquum* (Po), *P. dumosum* (Pd), *P. anonifolium* (Pa) and *P. reticulatum* (Pr), characterized by sesquiterpenes; (G3) P. *soledadense* (Ps), with monoterpenes and sesquiterpenes; and (G4) *P. sancti-felicis* (Psf) and *P. mituense* (Pm), characterized by phenylpropanoids. The essential oils in G2-4 showed important activity against *Botrytis cinereal* and were phytotoxic against *Lolium perenne*.

Considering the composition-based grouping of the EOs, we can conclude that the presence of bicyclogermacrene, 10-epi-Elemol, germacrene-D, caryophyllene, limonene, β-pinene, and/or apiol could be responsible for significant antifungal and herbicidal effects. β-pinene, apiol, and limonene showed antifungal activity, but not caryophyllene, suggesting that this compound could be a synergist.

These findings have important implications for the development of a *Piper* germplasm bank and the domestication of selected species to grant a sustainable biomass source for the production of essential oils with biopesticidal activity.

## Figures and Tables

**Figure 1 plants-11-01793-f001:**
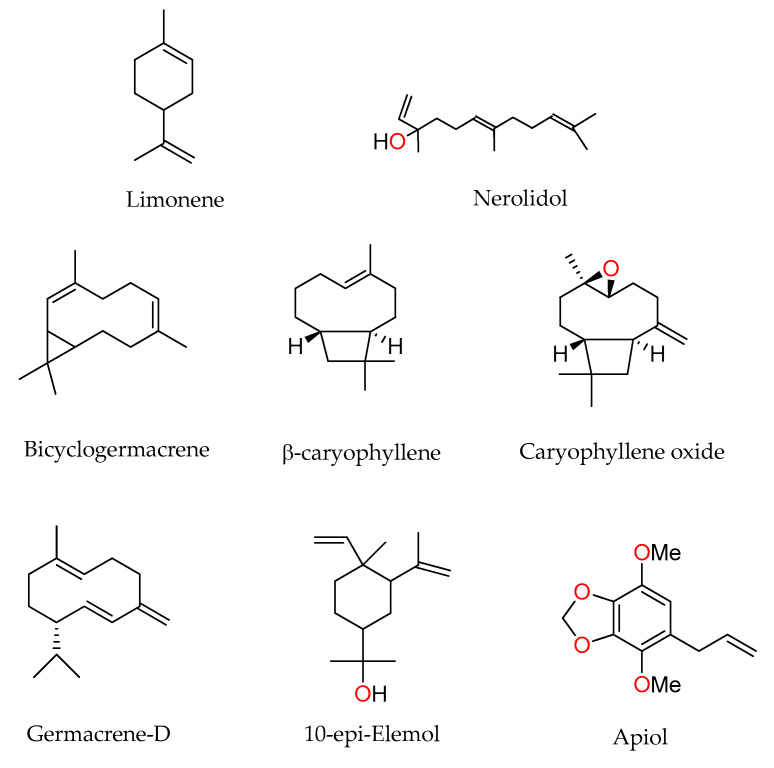
Main components of *Piper* essential oils.

**Figure 2 plants-11-01793-f002:**
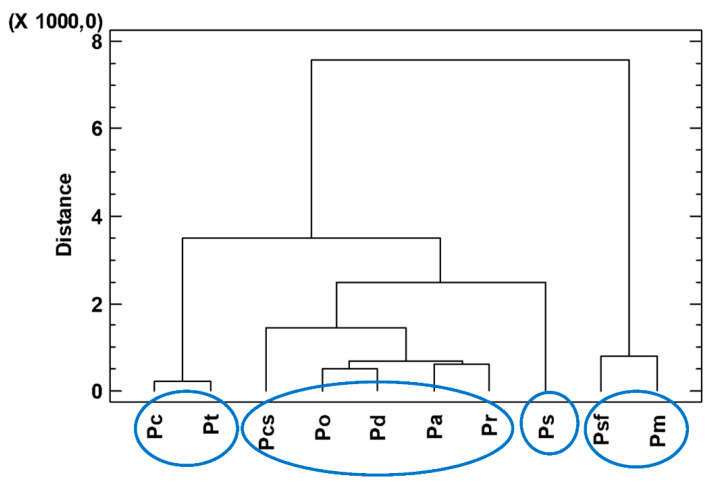
Dendrogram (Farthest Neighbor Method, Squared Euclidean) generated from cluster analysis of GC–MS data of ten *Piper* species from Peru. *P. coruscans* (Pc), *P. tuberculatum* (Pt), *P. casapiense* (Pcs), *P. obliquum* (Po), *P. dumosum* (Pd), *P. anonifolium* (Pa), *P. reticulatum* (Pr), *P. soledadense* (Ps), *P. sancti-felicis* (Psf), and *P. mituense* (Pm).

**Figure 3 plants-11-01793-f003:**
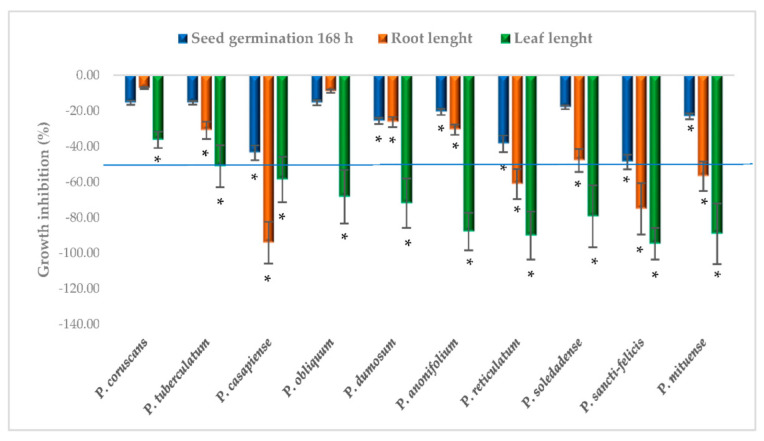
Phytotoxic effects (% inhibition) of the *Piper* EOs on *Lolium perenne,* seed germination, and root and leaf growth. Bars represent the average relative values ± standard error (*n* = 25 plants measured). * Significantly different from the control (*p* < 0.05), Fisher’s least significant differences test (LSD).

**Table 1 plants-11-01793-t001:** Chemical composition of the essential oils from *Piper coruscans* (Pc), *P. tuberculatum* (Pt), *P. casapiense* (Pcs), *P. obliquum* (Po), *P. dumosum* (Pd), P. *anonifolium* (Pa), *P. reticulatum* (Pr), *P. soledadense* (Ps), *P. sancti-felicis* (Psf), and *P. mituense* (Pm).

Compounds	R T	RIC	% Area									
			Pc	Pt	Pcs	Po	Pd	Pa	Pr	Ps	Psf	Pm
α-Pinene	3.82	937	0.5			6.0	2.0	0.7		1.9	0.4	0.3
Camphene	4.03	950				4.4	0.4					
β-Pinene	4.42	982	1.2			5.1	6.3	1.3		3.0	0.5	0.4
Limonene	5.22	1029	0.2			1.2	3.5	0.2	0.2	38.5	0.4	0.7
γ-Terpinene	5.74	1060					1.9				0.7	0.5
*cis*-Limonene oxide	7.21	1133								2.1		
D-Carvone	9.60	1244								2.5		
(-)-*cis*-β-Elemene	11.72	1339				0.6	2.6	1.0				1.0
α-Cubebene	12.03	1352	0.3		1.1	0.3	5.9	4.0	1.2	2.3		1.9
Copaene	12.61	1378	1.4	4.2	0.7	1.2	2.2	4.4	4.6	5.8	0.6	1.9
β-bourbonene	12.82	1388	0.6			0.4			0.7			
(-)-β-Copaene	12.90	1391						5.8		3.1		
β-Elemene	12.93	1393	1.1	3.3		0.6	4.5	0.6	2.5			2.1
α-Gurjunene	13.37	1413				1.3	0.3					
Caryophyllene	13.61	1424	8.0	9.7	4.7	6.3	6.8	11.3	4.5		4.1	2.9
(-)-β-Copaene isomer	13.79	1432	1.0	0.5		0.7	0.8	0.5	1.9	1.5		0.2
(+)-Aromadendrene	14.02	1443			1.1	1.2	0.3					
α-Humulene	14.35	1458	1.2	2.1	2.5	2.1	1.3	6.6	1.6		1.8	0.5
Neoalloocimene	14.51	1466				0.5	0.8	5.5				0.3
α-Amorphene	14.77	1478	1.8	0.8			0.6	1.1				0.2
Eudesma-3,7-(11)-diene	14.78	1479			0.7	1.4			4.0	5.8		
Germacrene-D	14.91	1484	4.0	5.0	1.0	3.5	10.4	9.6	12.6		1.6	6.7
(+)-β-Selinene	15.05	1491	4.9	0.6		0.4	0.5	2.6				
*trans*-α-Bergamotene	15.07	1492							4.3			1.4
107/93/121/189/133/79/81/91/109/147	15.22	1498	3.5	1.3								
Bicyclogermacrene	15.24	1500			2.2	7.9	16.5		8.1			9.0
161/105/81/204/119/162/134/91/159/131	15.32	1503						9.2				
β-Bisabolene	15.44	1509	33.4	40.2								
γ-Cadinene	15.59	1516	1.3		0.6	1.5	0.3		2.5	1.8		
Myristicin	15.67	1520					3.7		1.0			4.6
δ-Cadinene	15.76	1525	3.5	9.8	2.1	4.1	3.3	6.6	6.0		1.4	
*cis*-Calamenene	15.76	1525								1.8		
1,4-Cadinadiene	16.01	1537					2.1		0.8			1.0
10-epi-Elemol	16.31	1551	0.3			7.3						
Nerolidol	16.52	1561	10.2	4.5			0.6		1.4		0.6	
119/205/91/105/93/159/43/107/147/79	16.93	1581		0.5		6.3			0.9	4.9		0.7
Caryophyllene oxide	17.06	1587	2.0	2.6	10.2	0.4	0.6	4.4	1.2	8.4	1.9	
Guaiol	17.32	1600	1.6		2.9	3.1	0.9		0.9			
Humulene epoxide	17.59	1614			3.8	0.3		0.9		2.3	0.8	
Apiol	17.83	1626							15.0		76.1	51.6
α-Eudesmol	18.01	1635				4.5						
Muurola-4,9-diene	18.04	1637						2.5				
τ-Cadinol	18.20	1645							2.1			
τ-Muurolol	18.20	1645	0.8	1.3		3.1	0.6			1.6		0.8
Muurolol	18.28	1649	0.4						1.3	1.0		
105/93/91/161/119/79/133/159/81/77	18.30	1650			1.1							
Muurola-4,9-diene	18.32	1651						3.3				
β-Selinenol	18.41	1655	0.5			4.9						
95/121/161/204/43/109/105/81/164/108	18.46	1658			1.6	7.7	0.6					
Neointermedeol	18.48	1659		2.9								0.5
α-Cadinol	18.53	1661						1.9		1.6	0.3	
Bulnesol	18.72	1671	1.7			2.3						
Apiol isomer	18.91	1681					8.9		0.7			5.3
α-Bisabolol	18.98	1684	4.8									
93/91/79/133/105/119/107/189/81/67	19.49	1711			17.0							
93/133/91/105/79/107/119/77/106/121	21.13	1798			22.6							
93/133/91/105/79/107/119/77/106/121	21.55	1821			5.4							

RT, retention time (minutes). RIC, retention index on Teknokroma TRB (95%) dimethyl (5%) dimethylpolysiloxane (30 m × 0.25 mm ID and 0.25 µm phase thickness) column.

**Table 2 plants-11-01793-t002:** Overall composition (percent by chemical class) of the essential oils from *Piper coruscans* (Pc), *P. tuberculatum* (Pt), *P. casapiense* (Pcs), *P. obliquum* (Po), *P. dumosum* (Pd), P. *anonifolium* (Pa), *P. reticulatum* (Pr), *P. soledadense* (Ps), *P. sancti-felicis* (Psf), and *P. mituense* (Pm).

Chemical Class	Total (%)									
	Pc	Pt	Pcs	Po	Pd	Pa	Pr	Ps	Psf	Pm
Monoterpene hydrocarbons	1.9	0.0	0.0	16.7	14.1	2.2	0.2	45.5	2.0	1.9
Oxygenated monoterpenes	0.0	0.0	0.0	0.0	0.0	0.0	0.0	2.5	0.0	0.0
Sesquiterpene hydrocarbons	62.5	76.2	16.7	33.5	58.4	59.9	55.3	22.1	9.5	28.8
Oxygenated sesquiterpenes	12.1	6.8	16.9	25.9	2.1	7.2	5.5	14.9	3.0	1.3
Phenylpropanoids	0.0	0.0	0.0	0.0	12.6	0.0	16.7	0.0	76.1	61.5
Others	3.5	1.8	47.7	14.5	1.4	14.7	0.9	4.9	0.0	1.0
Total	80.0	84.8	81.3	90.6	88.6	84.0	78.6	89.9	90.6	94.5

**Table 3 plants-11-01793-t003:** Antifungal activity (% spore germination) against *Aspergillus niger*, *Botrytis cinerea,* and *Alternaria alternate*.

Species	Dose	Percent Spore Germination		
	(µg/mL)	*Aspergillus niger*	*Botrytis cinerea*	*Alternaria alternate*
*P. coruscans*	800	50 ± 5	51 ± 10	59 ± 9
	CI_50_ ^a^	>800	>800	>800
*P. tuberculatum*	800	77 ± 3	57 ± 9	87 ± 5
	CI_50_ ^a^	>800	>800	>800
*P. casapiense*	800	58 ± 3	24 ± 3	66 ± 4
	CI_50_ ^a^	>800	143.29 (115.1–178.3)	>800
*P. obliquum*	800	36 ± 3	15 ± 2	47 ± 5
	CI_50_ ^a^	>800	104.85 (84.9–129.4)	>800
*P. dumosum*	800	45 ± 2	28 ± 3	52 ± 5
	CI_50_ ^a^	>800	331.8 (282.6–389.4)	>800
*P. anonifolium*	800	62 ± 8	22 ± 3	59 ± 5
	CI_50_ ^a^	>800	110.16 (87.5–138.6)	>800
*P. reticulatum*	800	35 ± 3	26 ± 1	34 ± 4
	CI_50_ ^a^	>800	143.8 (104.3–198.4)	>800
*P. soledadense*	800	24 ± 2	15 ± 1	34 ± 5
	CI_50_ ^a^	298.5 (234.5–379.9)	129.9 (80.5–209.5)	>800
*P. sancti-felicis*	800	25 ± 1	25 ± 3	24 ± 1
	CI_50_ ^a^	410.3 (371.0–454.6)	202.7 (163.6–251.2)	317.2 (260.5–386.2)
*P. mituense*	800	39 ± 6	19 ± 1	32 ± 2
	CI_50_ ^a^	>800	191.4 (143.5–255.1)	>800
Thymol	CI_50_ ^a^	67.3 (63.0–71.7)	19.54 (22.94–15.74)	6.34 (17.68–2.04)

^a^ Dose needed to inhibit 50% of spore germination and 95% confidence limits (CL).

**Table 4 plants-11-01793-t004:** List of the plant species used and their origin (experimental field locations in Iquitos, Peru, and UTM coordinates), dry weight and essential oils yield.

Voucher Number	Plant Species	Origin, (UTM Coordinates)	Dry Weight (gr)	Essential Oils Yield (%)
039849	*P. coruscans*	Mazán District,	589.13	0.47
		710990; 9619525		
036367	*P. sancti-felicis*	Punchana District,	458.03	0.88
		695305; 9587673		
041044	*P. casapiense*	Mazán District,	422.69	0.13
		711556; 9623266		
027690	*P. obliquum*	Mazán District,	1239.67	0.13
		711396; 9623398		
042381	*P. anonifolium*	Mazán District,	1246.35	0.10
		711399; 9623398		
020115	*P. tuberculatum*	Mazán District,	565.39	0.13
		710947; 9619547		
040311	*P. dumosum*	San Juan Bautista District, 675962; 9559237	1102.49	0.078
042127	*P. reticulatum*	San Juan Bautista District, 676047; 9559417	766.39	1.26
033308	*P. soledadense*	San Juan Bautista District, 675915; 9559216	442.23	0.54
041473	*P. mituense*	San Juan Bautista District, 676010; 9559392	393.76	0.11

## Data Availability

Data available on request.

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
