# Peer review of "Antifungal and Herbicidal Potential of Piper Essential Oils from the Peruvian Amazonia"

_plants, 2022, doi:10.3390/plants11141793_

Round 1
Reviewer 1 Report
The authors carried out a study entitled "Antifungal and herbicidal potential of Piper essential oils from the Peruvian Amazonia".
The main compounds identified must appear in the abstract.
The keywords are not the same as the title.
The introduction should be further explored, see the state of the art carefully, in this way the authors can justify the reason for the study.
Table 1. A chemometric study can be used, a multivariate analysis of PCA and HCA, see example article https://www.mdpi.com/1420-3049/26/11/3462.
https://www.mdpi.com/1420-3049/26/23/7359
Table 1 Add the calculated and literature retention indices (RIC and RIL) also add the compound classes
4.3. Spore germination inhibition assay
Add Support References.
Add pictures of phytotoxic experiments,
Better describe the phytotoxic and fungicidal method, add supporting references
In conclusion, the authors should avoid due to data already discussed above.
Author Response
Dear Revisor.
Please see the attachment.
Thanks

Reviewer 2 Report
Dear Authors,
I had the opportunity to review your paper entitled "Antifungal and herbicidal potential of Piper essential oils from the Peruvian Amazonia", so after reading it a few times, I think that this paper is a scientific-relevant study and can be appropriate for publishing. On the other hand, some improvements need to be addressed in the manuscript, especially in the part of the discussion. I think that is a major thing to improve the discussion part and implement comparison with other similar research. Also, the statistical methodology in the manuscript is very basic and required an advanced level for publishing in the Plants.
There are my suggestions:
- avoid using "we" and "our" throughout the whole text; use passive form;
- reduce the number of words in the Abstract, highlighting only the most important results;
- check the way for showing references, I think it is necessary to use square brackets;
- line 58-65 all names of microorganisms need to be italic;
- line 116 - all names of microorganisms need to be italic;
- Table 2. better interpretation of results is required
- All Discussion part is not adequate; improve the discussion part and implement comparison with other similar research.
- Also, the statistical methodology in the manuscript is very basic and required an advanced level for publishing in the Plants.
Author Response
Estimado Revisor.
Consulte el archivo adjunto.
Gracias

Reviewer 3 Report
The purpose of the manuscript ID 1771726 was to analyze the chemical composition, fungicidal and phytotoxic activity of the essential oils of ten Peruvian Piper species. However, some minor comments and remarks should be taken into account before publishing in Plants:
In the introduction, please provide, based on a literature review, what dominant compounds of mono- and sesquiterpenes and phenylpropanoids are found in the essential oils of Piper species.
Please complete the activity results for the positive reference in the fungicidal and phytotoxic assays. It is possible to define the potency of the tested essential oils by comparing their results with reference compounds.
Author Response

(The authors gave the same response as above.)

Round 2
Reviewer 1 Report
The authors major reviews. Manuscript may be accepted for publication
Author Response
Dear Reviewer
We carry out a detailed review of the article.
Reviewer 2 Report
/
Author Response

(The authors gave the same response as above.)
